# Spike Protein Impairs Mitochondrial Function in Human Cardiomyocytes: Mechanisms Underlying Cardiac Injury in COVID-19

**DOI:** 10.3390/cells12060877

**Published:** 2023-03-11

**Authors:** Tin Van Huynh, Lekha Rethi, Ting-Wei Lee, Satoshi Higa, Yu-Hsun Kao, Yi-Jen Chen

**Affiliations:** 1International Ph.D. Program in Medicine, College of Medicine, Taipei Medical University, Taipei 11031, Taiwan; d142109010@tmu.edu.tw (T.V.H.); yjchen@tmu.edu.tw (Y.-J.C.); 2School of Biomedical Engineering, College of Biomedical Engineering, Taipei Medical University, Taipei 11031, Taiwan; lekhar@tmu.edu.tw; 3International Ph.D. Program for Biomedical Engineering, Taipei Medical University, Taipei 11031, Taiwan; 4Division of Endocrinology and Metabolism, Department of Internal Medicine, School of Medicine, College of Medicine, Taipei Medical University, Taipei 11031, Taiwan; b8801138@tmu.edu.tw; 5Division of Endocrinology and Metabolism, Department of Internal Medicine, Wan Fang Hospital, Taipei Medical University, Taipei 11696, Taiwan; 6Cardiac Electrophysiology and Pacing Laboratory, Division of Cardiovascular Medicine, Makiminato Central Hospital, Okinawa 901-2131, Japan; higa@haku-ai.or.jp; 7Graduate Institute of Clinical Medicine, College of Medicine, Taipei Medical University, Taipei 11031, Taiwan; 8Department of Medical Education and Research, Wan Fang Hospital, Taipei Medical University, Taipei 11031, Taiwan; 9Division of Cardiovascular Medicine, Department of Internal Medicine, Wan Fang Hospital, Taipei Medical University, Taipei 11031, Taiwan

**Keywords:** spike protein, cardiac dysfunction, mitochondrial dynamics, oxygen consumption rate, reactive oxygen species

## Abstract

Background: COVID-19 has a major impact on cardiovascular diseases and may lead to myocarditis or cardiac failure. The clove-like spike (S) protein of SARS-CoV-2 facilitates its transmission and pathogenesis. Cardiac mitochondria produce energy for key heart functions. We hypothesized that S1 would directly impair the functions of cardiomyocyte mitochondria, thus causing cardiac dysfunction. Methods: Through the Seahorse Mito Stress Test and real-time ATP rate assays, we explored the mitochondrial bioenergetics in human cardiomyocytes (AC16). The cells were treated without (control) or with S1 (1 nM) for 24, 48, and 72 h and we observed the mitochondrial morphology using transmission electron microscopy and confocal fluorescence microscopy. Western blotting, XRhod-1, and MitoSOX Red staining were performed to evaluate the expression of proteins related to energetic metabolism and relevant signaling cascades, mitochondrial Ca^2+^ levels, and ROS production. Results: The 24 h S1 treatment increased ATP production and mitochondrial respiration by increasing the expression of fatty-acid-transporting regulators and inducing more negative mitochondrial membrane potential (Δψm). The 72 h S1 treatment decreased mitochondrial respiration rates and Δψm, but increased levels of reactive oxygen species (ROS), mCa^2+^, and intracellular Ca^2+^. Electron microscopy revealed increased mitochondrial fragmentation/fission in AC16 cells treated for 72 h. The effects of S1 on ATP production were completely blocked by neutralizing ACE2 but not CD147 antibodies, and were partly attenuated by Mitotempo (1 µM). Conclusion: S1 might impair mitochondrial function in human cardiomyocytes by altering Δψm, mCa^2+^ overload, ROS accumulation, and mitochondrial dynamics via ACE2.

## 1. Introduction

Several variants of severe acute respiratory syndrome coronavirus 2 (SARS-CoV-2) are widespread across the globe. The spike (S) protein is important for the virus to enter host cells and initiate infection. Some COVID-19 vaccines generate neutralizing antibodies against the S protein, and may potentially reduce the severity of illness by blocking the interaction between the virus and its target cells. Although viral pneumonia is the predominant manifestation of SARS-CoV-2 infection, the infection is multisystemic. Cardiac injury is the second most prevalent manifestation of COVID-19 in hospitalized patients (20–30%) [1] and is associated with disease severity and associated mortality [1,2,3]. Cardiac injury includes acute complications and various post-acute sequelae, such as myocarditis, ischemic heart disease due to hypercoagulation [1], microvascular dysfunction, cardiac vasculitis of small vessels [4], arrhythmias, and heart failure. Viral infection causes either direct or indirect (systemic complications) cardiac injury. However, direct cardiac injury due to the presence of SARS-CoV-2 in the myocardium requires further confirmation [5,6].

The mechanisms underlying the damage and dysfunction of multiple organs in COVID-19 remain unknown. The S protein facilitates the expansion and remodeling of the hypertrophic layer (cardiac inflammation), which leads to various pathological cascades in the heart by binding the angiotensin-converting enzyme 2 (ACE2). The spike protein comprises subunits S1 and S2. S1 binds to ACE2 and mediates viral entry into host cells, thus, playing a key role in the pathogenesis of COVID-19. The S protein can cross [7] and disrupt the blood–brain barrier [8] and cause neurological deficits after binding to ACE2 [9,10,11]. The S protein, particularly S1, may directly target cardiac cells, leading to cardiac dysfunction. Avolio et al. demonstrated that SARS-CoV-2 rarely infected human cardiac pericytes through direct treatment. However, the treatment of human cardiac pericytes with S1, but not S2, induced several pathological alterations in these cells, which were mediated by the transmembrane glycoprotein CD147. The alterations included increased migration, decreased endothelial cell network formation capacity, increased proinflammatory cytokine levels, and apoptosis [12]. To the best of our knowledge, no study has focused on the effects of the S protein on cardiomyocytes. Compared with other organs, the heart requires a substantial amount of energy. Cardiac mitochondria occupy one third of the volume of adult cardiac muscle cells [13]. These vital organelles are crucial for cardiac function and other cellular processes. The reduction of the ATP synthesis capacity of cardiac mitochondria represents the key mechanism underlying cardiac contractile dysfunction [14,15]. They not only serve as the “power house” of cells, but also regulate cardiac ion homeostasis, cellular growth and apoptosis, protein quality, inflammation, and redox balance [16]. Therefore, in addition to the reduction of ATP synthesis, mitochondrial dysfunction impairs various functions, ultimately leading to cardiac diseases. In the present study, we investigated the effects of S1 on mitochondrial function in human cardiomyocytes.

## 2. Materials and Methods

### 2.1. Human Cardiomyocyte Culture

Human primary ventricular cardiomyocytes (AC16; Millipore, SCC109) were cultured in Dulbecco’s Modified Eagle Medium (DMEM)/Nutrient Mixture F-12 (Sigma, D6434) containing 2 mM L-glutamine (catalog no. TMS-002-C; EMD Millipore), 12.5% fetal bovine serum (catalog no. ES-009-B; EMD Millipore), and 1X penicillin–streptomycin solution (catalog no. TMS-AB2-C; EMD Millipore). AC16 cells (passages 7–9) were treated with two concentrations (0.1 and 1 nM) of S1 (Catalog no. Z03485; Genscript biotech, Piscataway, NJ, USA) with or without anti-ACE2 neutralizing antibody (Catalog no. 10108-MM37, Sino Biological Hong Kong, People’s Republic of China), anti-CD147 neutralizing antibody, and Mitotempo (GenScript Biotech, NJ, USA) for 24, 48, and 72 h (h). Untreated cells were used as the control.

### 2.2. Total ATP Assay

The whole-cell lysate was used to evaluate the levels of ATP using the ATPlite Luminescence Assay System, 300 Assay Kit (catalog no. 6016943; Perkin Elmer, Seer Green, UK), following the manufacturer’s instructions and normalized to total protein levels, as described previously [17].

### 2.3. Mitochondrial Bioenergetics

The Seahorse XFe24 extracellular flux analyzer (Seahorse Biosciences, North Billerica, MA, USA) was used to study mitochondrial bioenergetics. Control and S1-treated AC16 cells were seeded (50,000 cells/well) on Agilent seahorse XFe24 plates in XF DMEM medium (pH 7.4; 103575-100; Agilent Technologies) with 25 mM glucose, 1 mM pyruvate, and 2 mM glutamine. The mitochondrial bioenergetics were analyzed using the Seahorse XF Cell Mito Stress Test kit. The basal OCR was first measured, then consecutively treated with different metabolic stressors as indicted in tracing with oligomycin (1.5 µM, an ATP synthase inhibitor), carbonyl cyanide-4 (trifluoromethoxy) phenylhydrazone (FCCP, 2 µM, a mitochondrial uncoupler for producing maximal uncoupled respiration), and rotenone/antimycin A (1.5 µM, inhibitors for mitochondrial complex I/III). The seahorse XF real-time ATP rate assay kit was used to determine the total cellular ATP production rate and the fractional contributions from glycolysis and mitochondrial respiration with oligomycin (1.5 µM) and rotenone/antimycin A (1.5 µM), as described previously [18]. The ATP production rate was normalized to control group.

### 2.4. Mitochondrial Membrane Potential

The tetramethylrhodamine ethyl ester (TMRE) Mitochondrial Membrane Potential Assay Kit (catalog no. 701310; Cayman Chemical 1180 East Ellsworth Road Ann Arbor, MI, USA) was used to evaluate the mitochondrial membrane potential (Δψm) values of AC16 cells treated with S1 for 24 and 72 h. In brief, the cells were seeded on 96-well plates after 24 h (10,000 cells/well) and 72 h (5000 cells/well) treatments. The treated cells were subjected to TMRE (250 nM) staining for 30 min and then washed twice with the washing buffer provided in the kit. Hoechst (1 µg/mL) was used as the counterstain. The negative controls were treated with 20 µM FCCP before TMRE staining. The backgrounds of wells without any cells were also stained with TMRE. Fluorescence was measured using the ImageXpress Pico Automated Cell Imaging System (Molecular device, San Jose, CA, USA).

### 2.5. Mitochondrial Morphology

To evaluate the effects of S1 on mitochondrial fragmentation, MitoTracker™ Green FM (catalog no. M-7514; Thermo Fisher Scientific, Waltham, MA, USA) was used to stain the AC16 mitochondria after 24 and 72 h treatments; then, fluorescence images were obtained using the STELLARIS 8 confocal microscope (63 × oil lens; Leica Microsystem). The mitochondrial fragmentation count was determined using ImageJ. In addition, gross mitochondrial morphology was observed using transmission electron microscopy (TEM; catalog no. HT7700; Hitachi High-Technologies Corporation, Tokyo, Japan).

### 2.6. Immunoblotting

Proteins were extracted from AC16 cells using a mammalian protein extraction reagent containing protease and phosphatase inhibitors (Thermo Fisher Scientific, Waltham, MA, USA), as described previously [19]. The total protein (30 µg) was subjected to sodium dodecyl sulphate–polyacrylamide gel electrophoresis. The resultant bands were transferred onto polyvinylidene difluoride membranes, which were blocked using 5% bovine serum albumin and incubated overnight with primary antibodies against acetyl-coenzyme A carboxylase (ACC; catalog no. ab45174; Abcam), phosphorylated ACC (pACC; catalog no. #07303; Millipore), peroxisome-proliferator-activated receptor-gamma coactivator-1 (catalog no. ab106814; Abcam), CD36 (catalog no. sc-7309; Abcam), 5′ adenosine monophosphate-activated protein kinase (AMPKa)/phosphorylated AMPKa (pAMPKa; catalog nos. #2532/#2535; Cell Signaling), carnitine palmitoyl transferase 1 (CPT1; catalog no. ab104662; Abcam), CD147 (catalog no. sc-46700; Santa Cruz,), and ACE2 (catalog no. ab15348; Abcam). After washing the membranes five times (5 min each) using an orbital shaker, secondary antibodies were added and incubated at room temperature for 2 h. Bound antibodies were detected using an enhanced chemiluminescence detection system (Millipore, Darmstadt, Germany) and analyzed using AlphaEase FC (Alpha Innotech, San Leandro, CA, USA). Target bands were normalized to both glyceraldehyde-3-phosphate dehydrogenase (MBL, Nagoya, Japan) and control bands.

### 2.7. Mitochondrial Ca^2+^ and Reactive Oxygen Species Levels

X-Rhod-1/AM (Molecular Probes, Eugene, OR, USA) was used for staining mitochondrial Ca^2+^ (mCa^2+^) in AC16 cells. After 24 and 72 h treatments, the cells were incubated with 2.5 µM X-Rhod-1/AM in culture medium for 30 min in 5% CO_2_ at 37 °C. Next, 1 mM CoCl_2_-containing tyrode solution was added and incubated for 10 min. Then, the cells were stained with Hoechst 33342 (Sigma-Aldrich, St. Louis, MO, USA) for another 10 min before the images were obtained.

The reactive oxygen species (ROS) in the mitochondria were determined through MitoSOX Red (molecular probes) staining. AC16 cells were incubated with MitoSOX Red for 30 min in culture medium, which was followed by Hoechst staining for 10 min. Subsequently, fluorescent images were obtained.

Both X-Rhod-1 and MitoSOX Red fluorescence images were captured using fluorescence microscopy (EVOS M5000 Imaging System; Thermo Fisher Scientific). Fluorescence intensities were quantified using ImageJ and normalized to control intensities for statistical comparisons.

### 2.8. Intracellular Ca^2+^ Measurements

Fura-2 was used for measuring cytosol Ca^2+^ levels as described previously [19]. Briefly, AC16 cells treated with S1 for 24 h or 72 h were incubated with fura-2-acetoxymethyl ester in black-wall 96-well plates (5 μM; Life Technologies, Carlsbad, CA, USA) in F12-DMEM medium for 30 min at 37 °C and 5% CO_2_. Fura-2 fluorescence was read using a BMG multi-mode microplate reader (BMG lab tech, Allmendgrün 8, 77799 Ortenberg, Germany) with dual-excitation wavelengths of 340 and 380 nm. After measuring the baseline cytosol Ca^2+^ (cells were incubated in Ca^2+^-free Tyrode solution) for 2 min, thapsigargin (2.5 μM) was added to block endo/sarcoplasmic reticulum (SR) Ca^2+^-ATPase to evaluate Ca^2+^ release. As soon as the Ca^2+^ surge returned to the steady state, CaCl_2_ solution was added (2 mM) to study Ca^2+^ entry. The baselines and areas under the curves of F340/F380 fluorescence were used for comparing the baseline Ca^2+^ levels, SR-Ca^2+^ release, and Ca^2+^ entry between control cells and cells treated with S1.

### 2.9. Statistical Analysis

Continuous variables are expressed as mean ± standard error of mean values. Paired *t*-tests and one-way repeated-measures analysis of variance with post hoc Tukey’s tests were used to compare S1-treated and control cells. A *p* value of < 0.05 indicated statistical significance.

## 3. Results

### 3.1. S1 Increased Mitochondrial Respiration within 24 h but Impaired Mitochondrial Function within 72 h

AC16 cells treated with S1 for 24 h exhibited increased oxygen consumption rates (OCR) in both mitochondrial and glycolytic ATP production (Figure 1A); hence, increased OCR may be associated with total mitochondrial ATP synthesis. AC16 cells treated with S1 for 48 h and control cells showed similar OCRs in mitochondrial and glycolytic ATP synthesis. By contrast, AC16 cells treated for 72 h exhibited considerably lower OCR than control cells. These findings were consistent with the changes in whole-cell total ATP in AC16 cells after being treated with S1 (Appendix A).

The 24 h treatment increased basal and spare capacities, protein leak, nonmitochondrial respiration, and ATP-associated OCR (Figure 1B). By contrast, the 72 h treatment markedly decreased all parameters of mitochondrial respiration except proton leak (Figure 1B). We checked the effects of two different concentrations (0.1 nM and 1 nM) of S1 on AC16 mitochondrial real-time ATP production (Appendix A). However, only the 1 nM concentration (not the 0.1 nM concentration) had a significant effect (Appendix A); therefore, we only used 1 nM S1 as the final concentration for the remaining experiments.

### 3.2. S1 Increased Δψm within 24 h but Disrupted Δψm within 72 h

The mitochondrial membrane potential (Δψm) was evaluated using tetramethylrhodamine ethyl ester staining. The 24 h S1 treatment resulted in a higher negative Δψm compared with that in control cells (Figure 2A). This finding suggested the 24 h treatment temporally increased ATP synthesis with improved mitochondrial activity due to a more negative membrane potential. By contrast, the 72 h treatment resulted in a decrease in Δψm; this might be the reason why ATP synthesis was impaired in S1-treated AC16 cells (Figure 2B).

### 3.3. S1 Increased Cytosol and Mitochondrial Calcium Levels, and Induced Mitochondrial Fragmentation

Mitochondrial calcium (mCa^2+^) was evaluated using calcium indicator X-Rhod-1. Both the 24 h and 72 h (Figure 3A) treatments increased mCa^2+^ significantly compared to control groups. Furthermore, the 72 h S1 treatment also increased cytosol baseline Ca^2+^ levels, SR-Ca^2+^ release, and Ca^2+^ entry (Figure 3B).

Translocase of the outer membrane 20 (TOM20) plays an important role in transporting the essential proteins from the cytosol to mitochondria including the subunits of electron transport chain complexes, and proteins related to mitochondrial biogenesis [20,21]. We found that after being treated with S1 for 72 h, AC16 cells had lower TOM20 expression compared to control cells (Figure 4A). To study whether S1 may alter the mitochondrial dynamics of AC16 cells, leading to changed ATP production, we studied the mitochondrial fission and fusion via the MitoTracker staining of the mitochondria. Within 72 h, but not 24 h, S1 disrupted the mitochondrial networks presented in AC16 cells, leading to fragmentation and mitochondrial fission (Figure 4B).

### 3.4. S1 Increased the Expression of Fatty Acid Transporters within 24 h

Fatty acids are a key source of energy for ATP synthesis in cardiac mitochondria. The expression of proteins involved in fatty acid transport was investigated through immunoblotting. CD36 transports fatty acids from the plasma to cytosol. Acetyl-CoA carboxylase (ACC) catalyzed the synthesis of malonyl-CoA, which is an inhibitor of the fatty acyl-CoA transporters CPT1. S1 treatment for 24 h increased the phosphorylation of both AMPK and ACC (Figure 5A), which inhibited the activity of ACC and promoted that of malonyl-CoA decarboxylase (MCD) to catalyze the conversion of malonyl-CoA to acetyl-CoA. However, S1 did not change the expression of peroxisome proliferator-activated receptor gamma coactivator 1-alpha (PGC-1α), which plays a role as the critical mediator in pathways of mitochondrial biogenesis. Finally, the transporter levels of fatty acids and their metabolites to mitochondria were increased, which explains the increased ATP synthesis after the 24 h S1 treatment (Figure 5A). Notably, the 72 h treatment of S1 did not alter the expression of the proteins involved in fatty acid transport (Figure 5B) or that of the ACE2 and CD147 of the SARS-CoV-2 receptors (Figure 5C).

### 3.5. S1 Effect Blocked by ACE2 but Not CD147 Neutralization Antibodies

Previous studies have found that ACE2 and CD147 can be the receptors mediating S1 and SARS-CoV-2 cell entry. Neutralizing ACE2 using an anti-ACE2 antibody blocked the effects of S1 on AC16 OCR (Figure 6A). However, the anti-CD147 antibody did not change the decrease in glycolysis and mitochondrial OCR after treatment with S1 for 72 h (Figure 6B).

### 3.6. S1 Increased Mitochondrial ROS Production within 72 h but Not within 24 h

ROS are the product of oxidative phosphorylation in mitochondria during respiration. They are involved in various pathological pathways in cardiomyocytes, such as mitochondrial dysfunction, fission, fragmentation, autophagy, and apoptosis. The accumulation of ROS also induces fatal cardiac arrythmias. S1 increased ROS levels in AC16 cells within 72 h but not within 24 h (Figure 7A). Moreover, being pretreated with Mitotempo (1 μM) for 30 min before adding S1 partly attenuated the impact of S1 on both glycolysis and mitochondrial ATP OCR (Figure 7B).

## 4. Discussion

Cardiac injury in the setting of COVID-19 results from either asystemic response to the infection or direct damage caused by the virus. However, further evidence is needed to confirm the presence of SARS-CoV-2 in the myocardium. Viral proteins may be distributed intracellularly and target various organelles, particularly mitochondria and the sarcoplasmic reticulum, in human-induced pluripotent stem cell-derived cardiomyocytes; the virus does not infect these cells to alter their function and viability [22]. The nonstructural proteins of SARS-CoV-2 bind to the mitochondrial permeability transition pore and lead to altered mitochondrial morphology and mitochondrial dysfunction by interacting with the coiled-coil domain containing 58 peripheral blood mononuclear cells [22]. S1, which acts as a ligand in the binding of the virus to host ACE2 and mediates viral entry into host cells, may induce pathophysiological changes in brain endothelial cells by altering the delivery of molecules and functions of mitochondria [9], disrupting the integrity of the blood–brain barrier through Rho activation [8], or impairing the functions of human cardiac pericytes through CD147-mediated activation of ERK1/2 [12]. Moreover, a relatively higher incidence of myocarditis in adolescents after receiving COVID-19 mRNA vaccines (encoding S protein) also raises this possibility of direct S protein involvement in myocardial injury [23,24]. To the best of our knowledge, the present study is the first to report that a short-term (24 h) treatment with S1 promotes ATP synthesis, increases basal OCR capacity, and preserves OCR capacity in AC16 cells. These adaptive responses help cells tolerate the S1-induced stress. Many viruses use the host cellular metabolic machinery to synthesize the energy required for viral replication by modulating ion channels [25,26]; viral proteins may form a hydrophobic pore to facilitate ion transport across the cell membrane [27]. Thus, the short-term effect of S1 on ATP synthesis may contribute to happy hypoxemia in COVID-19, in which patients exhibit severe hypoxemia without any sign of respiratory distress [28]. However, these effects were not maintained after 48 or 72 h. By contrast, substantial decreases were noted in ATP synthesis; basal, maximal, and spare OCR capacities; and nonmitochondrial respiration. The 24 h treatment increased proton leakage, which might have led to the accumulation of ROS [29,30,31,32] and impaired mitochondrial function during prolonged treatment (72 h). The energy required for the contraction of cardiomyocytes is generated primarily through mitochondrial oxidative phosphorylation. Thus, decreased mitochondrial ATP synthesis impairs cardiac function. Using human brain microvascular endothelial cells, a study revealed that the S1 decreased mitochondrial respiration [9].

In the heart, fatty acids are the primary sources of key substrates for ATP synthesis, accounting for 60–80% of total cardiac ATP synthesis [33]. ATP synthesis is stringently regulated by several mechanisms, including the uptake of fatty acids from the plasma into the cells by CD36/fatty acid translocase, the inhibition of CPT1 by malonyl-CoA, which is synthesized by the conversion of acetyl-CoA by ACC (inactivated by AMPKa), and the degradation of malonyl-CoA by MCD [34]. ACC catalyzes the synthesis of malonyl-CoA to inhibit the CPT1-mediated transport of acyl-CoA (a substrate for the beta-oxidation of fatty acids) from the cytosol to mitochondria. Notably, the 24 h treatment increased the phosphorylation of AMPKa, which inactivates ACC activity and activates MCD, which converts malonyl-CoA into acetyl-CoA. Therefore, activated AMPKa promotes the uptake of fatty acid substrates into mitochondria for oxidative phosphorylation via phosphorylation of ACC (inactivated form). However, these effects of S1 were temporary and did not last for 72 h. This finding explains why ATP synthesis and mitochondrial respiration increased in AC16 cells after the 24 h treatment. Interestingly, though PGC-1α phosphorylation or expression levels were not altered, decreased expression of TOM20 could impair the transport of mitochondrial transcription factor A and B, disrupting mitochondrial biogenesis and, hence, mitochondrial dysfunction. Even though we did not measure the ETC complex activities, TOM20 has an essential role in transporting the subunits of these complexes, particularly complex I [20,35], as well as in mitophagy [36]. Decreased TOM20 impairs complex I activity and mitophagy, resulting in decreased mitochondrial function.

Ca^2+^ is involved in various cellular regulatory mechanisms. mCa^2+^ exerts biphasic effects on energetics, cardiac function, and oxidative balance. Low mCa^2+^ levels may result in the tricarboxylic acid cycle failing to generate enough reduced fuel substrates to match the rate of oxidation in the mitochondrial electron transport chain (ETC), leading to a redox imbalance. Increased levels of mCa^2+^ promote mitochondrial oxidative phosphorylation by activating pyruvate dehydrogenase, α-ketoglutarate dehydrogenase, isocitrate dehydrogenase, complex III, and ATP synthase. However, an mCa^2+^ overload leads to the excessive production of reducing equivalents and redundant delivery of electrons to ETC, which promotes ROS production [37,38]. Furthermore, excessive mCa^2+^ may induce the formation of mitochondrial permeability transition pores, thus, disrupting Δψm and causing mitochondrial dysfunction [39,40]. In this study, we found that S1 increased mCa^2+^ in AC16 cells after treatment for 24 h and maintained high mCa^2+^ in the 72 h treatment. These results may have arisen from the effects of increased SR-Ca^2+^ release as well as Ca^2+^ entry by S1.

The activity of ATP synthase (complex) is maintained by the proton (H^+^)-driving force, which is a combination of Δψm and the H^+^ gradient. Thus, any change in Δψm impairs mitochondrial function. We evaluated TMRE fluorescence to determine the Δψm of AC16 cells. The 24 h treatment increased Δψm, enhancing the activity of ATP synthase and, thus, increasing ATP synthesis and mitochondrial oxygen consumption. However, the 72 h treatment considerably disrupted Δψm. These findings partly explained the decreases in mitochondrial ATP synthesis and oxygen consumption after the 72 h treatment. Δψm is directly associated with mitochondrial ROS (mROS) production. It exerts a biphasic effect on mROS production, as postulated by the redox-optimized ROS balance hypothesis. In the range of optimized Δψm, mROS production is maintained at physiological levels; extreme high or low Δψm may induce the overproduction of mROS. Through MitoSOX Red staining, we found that AC16 cells treated with S1 for 72 h exhibited higher levels of ROS than control cells; this might have been because S1 substantially disrupted Δψm and increased mROS production. In turn, the accumulation of ROS exacerbated mitochondrial dysfunction. The Mitotempo pretreatment could partly attenuate the impact of S1 on mitochondrial respiration. This result further confirmed the role of ROS in mitochondrial dysfunction, at least in part. Accordingly, mitochondrial antioxidants are the potential agents to reduce the detrimental effects of S1 by oxidative stress.

Mitochondrial dynamics, including morphological changes (fragmentation) and mitochondrial fission, are critical for maintaining mitochondrial function. In a study using human-induced pluripotent stem cell-derived cardiomyocytes, a nonstructural protein of SARS-CoV-2 (namely, M-protein) caused mitochondrial fragmentation and altered mitochondrial bioenergetics [22]. Using MitoTracker confocal fluorescence microscopy, we observed that mitochondrial fragmentation and network disruption increased in S1-treated (72 h) AC16 cells compared with control cells. Furthermore, TEM revealed that the S1 treatment for 72 h induced mitochondrial fission in AC16 cells, as evidenced by the decrease in size and increase in the number of mitochondria compared with the findings for control cells. Changes in mitochondrial shape and morphology might have resulted from not only mitochondrial dysfunction and excessive ROS accumulation, but also impaired mitochondrial function. Using murine cardiomyocytes, a study revealed that coxsackievirus B3 (CVB3) exerted similar effects on mitochondrial fragmentation by activating dynamin-related protein 1 (Drp1); the inhibition of Drp1 prevented mitochondrial damage and myocardial injury in CVB3-induced myocarditis [41]. Our findings suggested that S protein-induced myocardial injury may be prevented using 1-methyl-4-phenyl-1,2,3,6-tetrahydropyridine blocking agents, such as cyclosporin, or through targeting mitochondrial fission proteins, such as Drp1, to reduce mitochondrial fragmentation and fission. In addition, cardiomyocytes may be protected by restoring the redox balance, which prevents the induction of detrimental signaling cascades.

A previous study had proved that ACE2, which plays an important role in SARS-CoV-2 cell entry, is highly expressed in the human heart [42,43]. Moreover, in a recent study, by blocking ACE2 and CD147, Avolio et al. found that the effects of the spike protein on human cardiac pericytes were mediated via CD147 but not ACE2 [12]. Furthermore, the S1 effect was blocked by ACE2 but not CD147 neutralization antibodies in AC16 cells, suggesting that S1 may dysregulate mitochondrial activity via ACE2. Therefore, blocking ACE2 signaling might completely abolish the effects of S1 on mitochondrial activity, ROS, and ATP production in cardiomyocytes. The binding of S1 to the ACE2 receptor initiates the cleavage of ACE2 at the ectodomain and endodomain through the host’s metallopeptidase domain 17 and transmembrane protease serine 2. This process leads to the shedding and downregulation of ACE2 [44,45]. ACE2 acts as a vital counter-regulatory enzyme of the renin–angiotensin axis by converting Ang II to Ang 1–7. Accordingly, Ang 1–7 activates the MAS/G protein-coupled receptor to provide the protective counteracting effect of Ang II. Therefore, SARS-CoV-2 infection decreases ACE2 on the cell membrane and their activity, and increases Ang II-mediated pathophysiology. In this study, we also checked the protein expressions of CD147 and ACE2, and showed that AC16 cells expressed both receptors and their expression levels were not changed after treatment with S1. However, we did not evaluate the cardiac expression of both receptors in animal models.

In summary, as shown in Figure 8, although S1 increased cardiomyocyte mitochondrial respiration by facilitating the transport and uptake of fatty acids as well as increasing the Δψm in short term, longer treatment with S1 impaired the cardiomyocyte mitochondrial respiration, leading to the accumulation of ROS by increasing mCa^2+^ and disrupting Δψm. ACE2, but not CD147, is suggested to mediate the internalization and effects of S1.

### 4.1. Limitations

Our study has some limitations. We could not elucidate the complete mechanisms underlying S1-induced mitochondrial dysfunction. Furthermore, the concentration of S1 used in our study may differ from that in the cells of patients with COVID-19. Moreover, the effects of S1 may be moderated by various factors in the complex disease environment involving multiple organs. Thus, whether our findings are relevant in a real-world clinical setting requires further investigation. The role of S1 should be investigated using an animal model to evaluate its systemic effects and associated immune responses.

### 4.2. Conclusions and Future Perspective

To the best of our knowledge, this is the first study to evaluate the effects of S1 on mitochondrial function in human cardiomyocytes. Although S1 improved mitochondrial function in AC16 cells in the short term, prolonged S1 treatment led to mitochondrial dysfunction due to the disruption of Δψm, mitochondrial Ca^2+^ overload, accumulation of ROS, and alteration of mitochondrial dynamics. These effects of subunit S1 of the S protein translate into irreversible cardiac remodeling.

## Figures and Tables

**Figure 1 cells-12-00877-f001:**
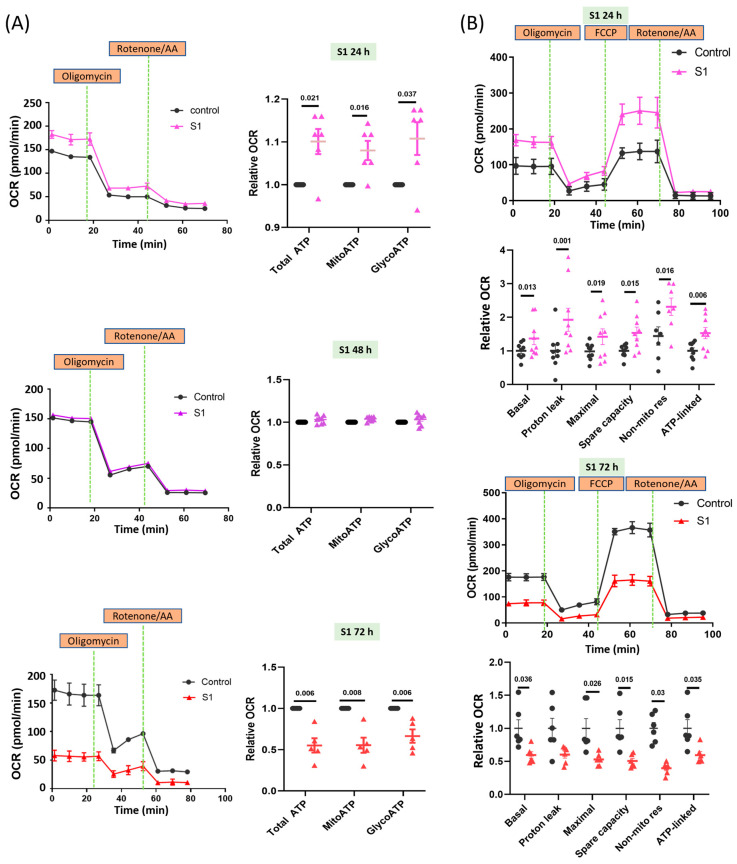
Effects of S1 treatment at different durations on ATP synthesis in AC16 cells. (**A**) Seahorse real-time ATP rate assay showed that ATP production in AC16 cells was increased at 24 h, but reduced at 72 h after S1 (1 nM) treatment (*n* = 6 independent experiments for 24 h, *n* = 8 independent experiments for 48 h, and *n* = 5 independent experiments for 72 h). Total ATP production rate is the sum of the ATP production rate from mitochondrial oxidative phosphorylation and glycolysis. (**B**) Seahorse Mito Stress assay showed that mitochondrial bioenergetics in AC16 cells were increased at 24 h, but decreased at 72 h after S1 (1 nM) treatment (*n* = 9 independent experiments for 24 h and *n* = 6 independent experiment for 72 h). Paired *t*-test was performed for statistical analysis; *p* ≤ 0.05 indicated statistical significance.

**Figure 2 cells-12-00877-f002:**
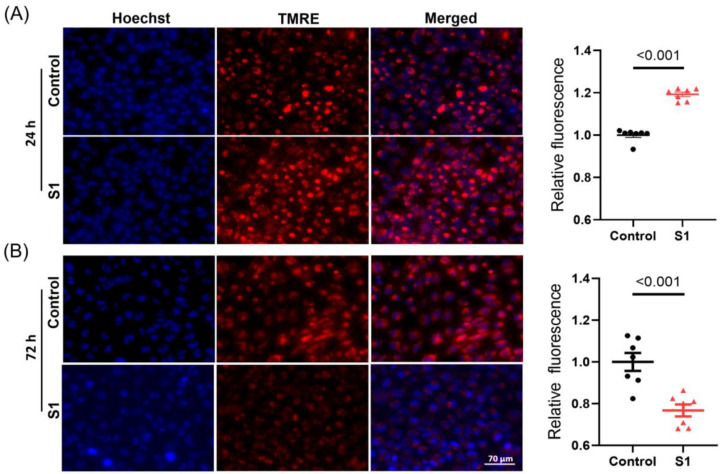
AC16 mitochondrial membrane potential was evaluated using TMRE staining after 24 h and 72 h S1 treatment. S1 increased AC16 mitochondrial membrane potential after 24 h treatment compared to the control group (**A**). However, after 72 h treatment, S1 decreased mitochondrial membrane potential in AC16 cells (**B**). *n* = 7 independent experiments. Paired *t*-test was used for comparing the difference in TMRE fluorescence between groups. TMRE: tetrametylrhodamine ethyl ester.

**Figure 3 cells-12-00877-f003:**
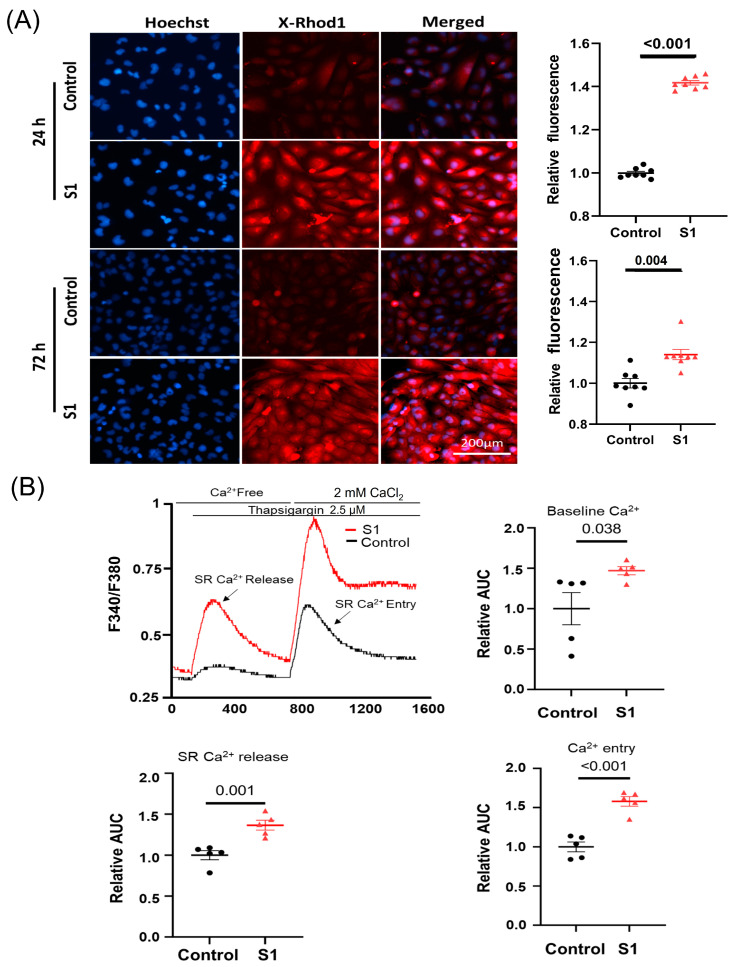
The effects of S1 on AC16 intracellular and mitochondrial calcium (mCa^2+^) levels at 24 h and 72 h post-treatment using Fura-2 and X-Rhod-1 staining, respectively. (**A**) S1 protein increased mCa^2+^ levels in both 24 h- and 72 h-treated cells compared to control groups (*n* = 8 independent experiments). (**B**) S1 protein increased baseline cytosol Ca^2+^ level, SR-Ca^2+^ release, and Ca^2+^ entry (*n* = 5 independent experiments) after treatment for 72 h in AC16. Paired *t*-test was used for comparing control and treated cells.

**Figure 4 cells-12-00877-f004:**
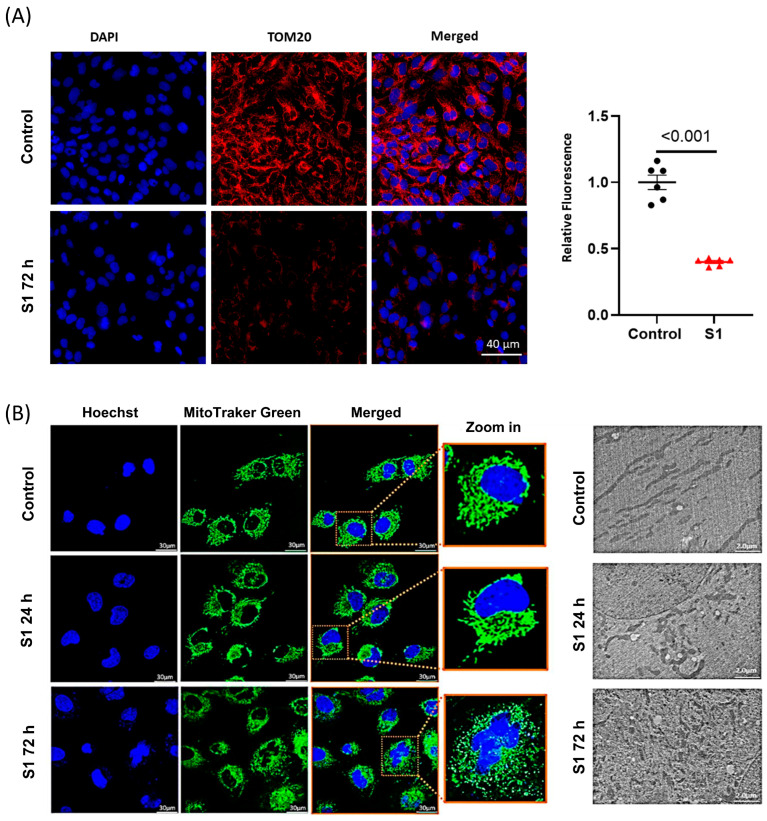
The effects of S1 on mitochondrial dynamics. (**A**) S1 treatment for 72 h decreased TOM20 expression, which is important in transporting essential proteins from cytosol to mitochondria including mitochondrial transcription factors and ETC complex subunits (*n* = 6 independent experiments). (**B**) Mitochondrial morphology evaluated using confocal fluorescence microscopy and transmission electron microscopy. S1 protein disrupted mitochondrial networks and led to mitochondrial fragmentation and mitochondrial fission within 72 h but not within 24 h (*n* = 50 different cells were evaluated for treatment and control).

**Figure 5 cells-12-00877-f005:**
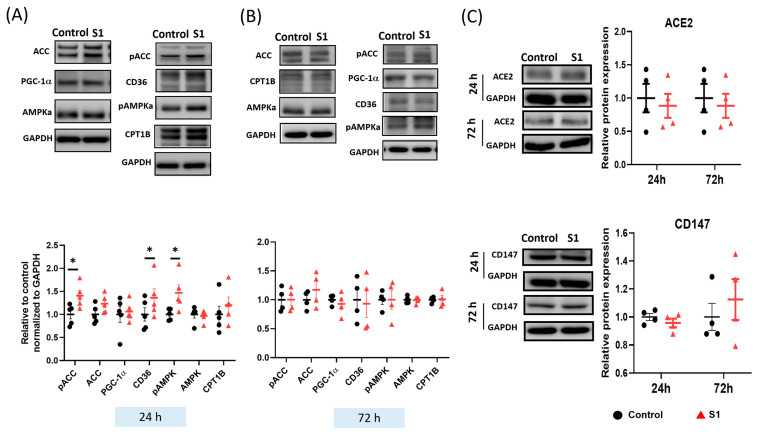
Effects of S1 on the expression of fatty acid transporters and spike protein receptors in AC16 cells after 24 and 72 h treatments. (**A**) S1 (24 h) increased the protein expression of CD36 and phosphorylation of acetyl-coenzyme A carboxylase (ACC) and 5′-adenosine monophosphate–activated protein kinase catalytic subunit alpha-2 (AMPKa), but not the expression of peroxisome proliferator-activated receptor-gamma coactivator-1 and carnitine palmitoyl transferase 1B, total AMPKa, and ACC (*n* = 4–6 independent experiments). (**B**) S1 did not alter the expression of fatty acid transporters at 72 h (*n* = 4 independent experiments). (**C**) S1 did not change protein expression of CD147 and ACE2 of the SARS-CoV-2 receptors at 24 h and 72 h. *n* = 4–6 independent experiments. Paired *t*-test was performed to compare the S1 protein-treated and control cells: * *p* ≤ 0.05.

**Figure 6 cells-12-00877-f006:**
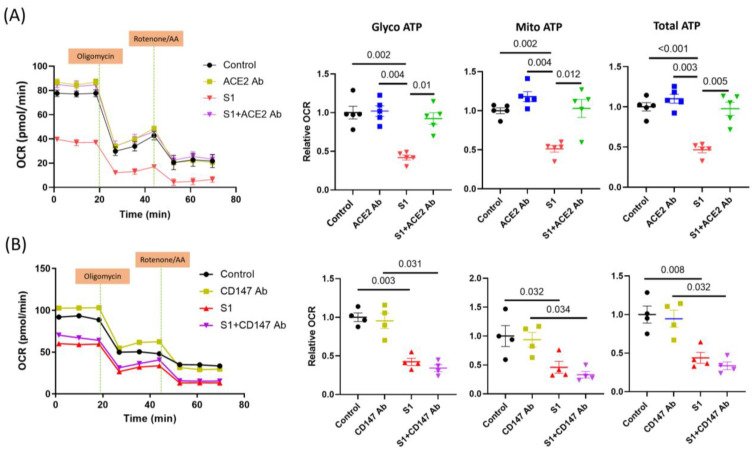
Effects of S1 protein on AC16 mitochondrial ATP respiration in presence of anti-CD147 and anti-ACE2 neutralizing antibodies. Anti-ACE2 antibody (*n* = 5 independent experiments), but not anti-CD147 (*n* = 4 independent experiments), blocked the effects of S1 (72 h) on mitochondrial ATP production in AC16 cells. One-way ANOVA followed by Tukey post-hoc analysis was used for comparing the difference between groups.

**Figure 7 cells-12-00877-f007:**
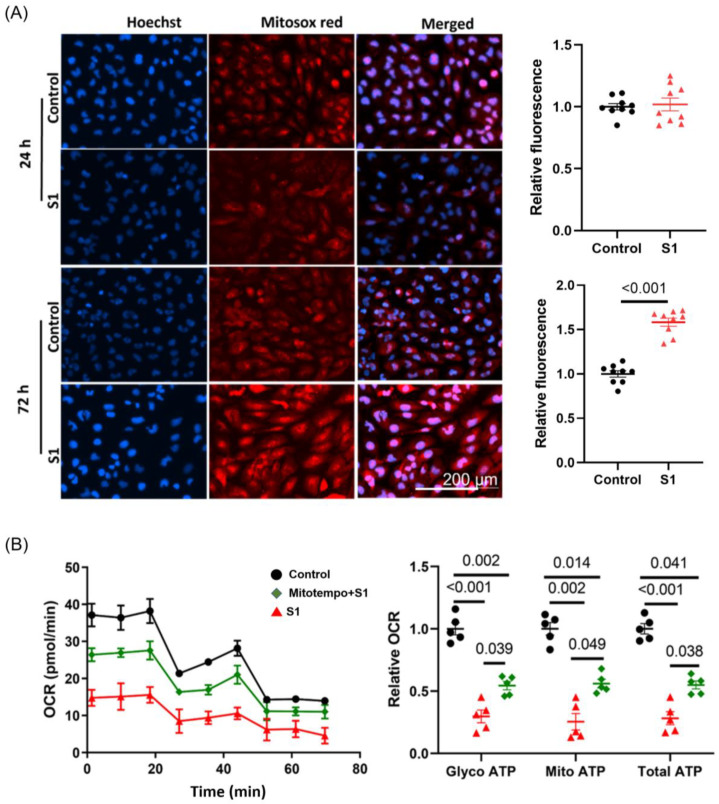
Effects of S1 protein on mitochondrial ROS and Mitotempo on ATP production in AC16. (**A**) ROS production (evaluated using MitoSOX Red fluorescence) was increased in AC16 cells after treatment with S1 for 72 h, but not 24 h. Paired *t*-test was performed to compare S1 protein-treated and control cells (*n* = 9 independent experiments). (**B**) Mitotempo (1 μM) pretreatment partly attenuated the effects of S1 (72 h) on AC16 ATP production. One-way repeated ANOVA followed by Tukey post-hoc analysis was used for comparing the effects of S1 and Mitotempo on AC16 ATP production (*n* = 5 independent experiments). ROS: reactive oxygen species.

**Figure 8 cells-12-00877-f008:**
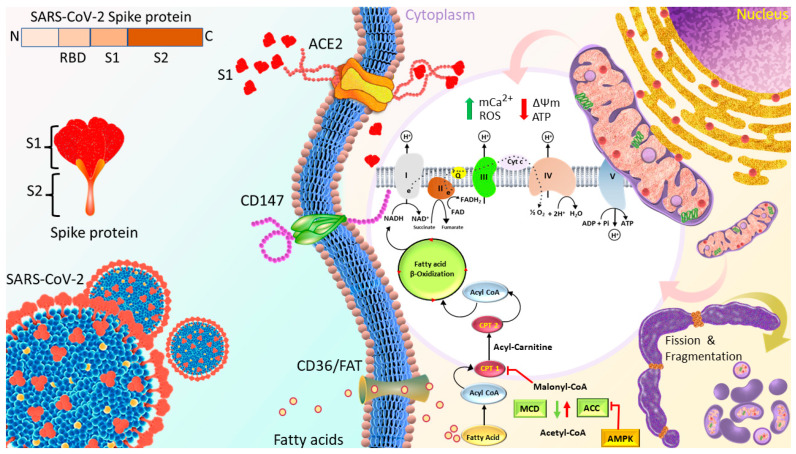
Proposed mechanisms through which S1 induced cardiac mitochondrial dysfunction which leads to cardiac injury in COVID-19 patients. Spike protein is the glycosylated protein that covers the surface of SARS-CoV-2 and binds to the host ACE2 receptor to mediate the viral cell entry. It is composed of S1 and S2 subunits that are responsible for ACE2 binding and membrane fusion, respectively. S1 possibly binds to ACE2 on the AC16 membrane, and is then internalized into the cytosol and localized in organelles, such as mitochondria, which induces the transient increase in fatty acids transport and uptake for biogenetics, Δψm, and permanent mCa^2+^, and disrupts Δψm later, finally impairing mitochondrial function and promoting ROS production. In turn, ROS further exacerbates mitochondrial function and mitochondrial fragmentation. Moreover, S1 also causes downregulation of TOM20; this effect might inhibit the pathways leading to mitochondrial biogenesis. ACE2, angiotensin converting enzyme 2; FAT, fatty acid translocase; PCT1/2, Carnitine palmitoyltransferase 1/2; MCD, Malonyl-CoA Decarboxylase; ACC, acetyl-CoA carboxylase; AMPK, AMP-activated protein kinase; mCa^2+^, mitochondrial Calcium, Δψm, mitochondrial membrane potential; ROS, reactive oxygen species.

## Data Availability

The datasets generated during or analyzed during the current study are available from the corresponding author on reasonable request.

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
