# Peer review of "Spike Protein Impairs Mitochondrial Function in Human Cardiomyocytes: Mechanisms Underlying Cardiac Injury in COVID-19"

_cells, 2023, doi:10.3390/cells12060877_

Round 1

Reviewer 1 Report

The manuscript entitled "Spike Protein Impairs Mitochondrial Function in Human Cardiomyocytes: Mechanisms Underlying Cardiac Injury in COVID-19” by Huynh T.V. et al. reported that mitochondrial bioenergetics in cardiomyocytes is impaired by the spike protein. The title matches well with the content and the work is well described. Methods are clearly stated and easily replicable. Results and conclusions are correct and clearly supported by the interpretation of the data. 

Author Response

Response to the Reviewer 1

Thank you very much for your detailed comments. These comments were very instructive and helpful for improving our manuscript and for our future research. We highly appreciate your careful and detailed review of our manuscript. The responses to these comments are enumerated below:

  1. The manuscript entitled "Spike Protein Impairs Mitochondrial Function in Human Cardiomyocytes: Mechanisms Underlying Cardiac Injury in COVID-19” by Huynh T.V. et al. reported that mitochondrial bioenergetics in cardiomyocytes is impaired by the spike protein. The title matches well with the content and the work is well described. Methods are clearly stated and easily replicable. Results and conclusions are correct and clearly supported by the interpretation of the data.

Thank you for your encouragement. We highly appreciate your careful and detailed review of our manuscript.

  1. English language and style are fine/minor spell check required.

We appreciate this comment very much. According to your suggestion, we have made spelling by a native English speaker.

The above descriptions are the responses to your comments and suggestions.

Sincerely yours,

Yu-Hsun Kao, PhD

Graduate Institute of Clinical Medicine, Taipei Medical University

Reviewer 2 Report

Minor comments:

Keywords - there is no hint of damage in the keywords - perhaps this should be addressed

p2 Line 50. 'Some vaccines neutralize the antibodies against....' I don't get what is being said here - does this need a massive re-wording?

p2 LIne 65. Given the role of ACE2 RECEPTOR in S-protein cell entry, there is a potential confusion here. I presume the activation of the ACE2 ENZYME is a direct result of the infection rather than due to a direct interaction with S2.

If both the enzyme and the receptor need to be discussed then the authors cannot simply refer to ACE2 without saying which one

General: Generally well written, but some mistakes due to non-native English speakers, especially line 326 +/- ~5 lines. Could be fixed by the publishers?

Figure 1. y-axis not labelled in same cases; legend insufficiency clear to interpret figures. In this and other figures (e.g. Figure 5) the edge has been cropped.

p12 line 280. The discussion in

Cardiovascular Manifestation of the BNT162b2 mRNA COVID-19 Vaccine in Adolescents, Suyanee Mansanguan et al.

https://doi.org/10.3390/tropicalmed7080196 

also raises htis possibility of direct spike involvement.

Author Response

Thank you very much for your detailed comments. These comments were very instructive and helpful for improving our manuscript and for our future research. We highly appreciate your careful and detailed review of our manuscript. The responses to these comments are enumerated below:

  1. Keywords - there is no hint of damage in the keywords - perhaps this should be addressed

We agree that the “hint of damage” should be mentioned in the keywords, we have replaced “human cardiomyocytes” with “cardiac dysfunction” in the revised manuscript.

  1. p2 Line 50. 'Some vaccines neutralize the antibodies against....' I don't get what is being said here - does this need a massive re-wording?

We are sorry for this unclear sentence. We have revised it as follows “Some COVID-19 vaccines generate neutralizing antibodies against the S protein, and may potentially reduce the severity of illness by blocking the interaction between the virus and its target cells (page 2, lines 50-52, red font).”    

  1. p2 Line 65. Given the role of ACE2 receptor in S-protein cell entry, there is a potential confusion here. I presume the activation of the ACE2 enzyme is a direct result of the infection rather than due to a direct interaction with S2. If both the enzyme and the receptor need to be discussed then the authors cannot simply refer to ACE2 without saying which one.

We are sorry for this incorrect presentation about the “activation of the ACE2 enzyme”. S1 binds ACE2 for viral entry but not activates ACE2 activity. We have corrected this mistake in the revised Introduction as follows “S protein facilitates the expansion and remodeling of the hypertrophic layer (cardiac inflammation), which leads to various pathological cascades in the heart by binding the angiotensin-converting enzyme 2 (ACE2)“. In addition, we agree with your comment that we should discuss both the enzyme and the receptor function for ACE2.  Binding of S1 to ACE2 receptor initiates the cleavage of ACE2 at ectodomain and endodomain by host's metallopeptidase domain 17 and transmembrane protease serine 2. This process leads to shedding and downregulation of ACE2 (Tipnis SR., et al., Cell 2020; Ou X, et al., Nat Commun 2020). ACE2 acts as a vital counter-regulatory enzyme of the renin-angiotensin axis by converting Ang II to Ang 1-7. Accordingly, Ang 1-7 activates MAS/G protein coupled receptor to provide protective counteracted effect of Ang II. Therefore, SARS-CoV-2 infection decreases ACE2 on cell membrane and their activity and increases Ang II-mediated pathophysiology.

In this study we also checked the protein expressions of CD147 and ACE2, and showed that AC16 cells expressed both receptors and their expression levels were not changed after treatment with S1. How-ever, we did not evaluate cardiac expression of both receptors in animal models. According to your suggestion, we have commented this important issue in the revised Discussion (page 14, lines 397-407, red font).

  1. General: Generally well written, but some mistakes due to non-native English speakers, especially line 326 +/- ~5 lines. Could be fixed by the publishers?

Thank you for your detail and helpful comment. We have made English editing by native English speaker. The changes were highlighted in red font, especially lines 330-341.

  1. Figure 1. y-axis not labelled in same cases; legend insufficiency clear to interpret figures. In this and other figures (e.g. Figure 5) the edge has been cropped.

We apology for these mistakes. We have checked and revised Figures. We also improve the Figure legends to describe more clearly in the revised manuscript.

  1. p12 line 280. The discussion in Cardiovascular Manifestation of the BNT162b2 mRNA COVID-19 Vaccine in Adolescents, Suyanee Mansanguan et al. (https://doi.org/10.3390/tropicalmed7080196) also raises this possibility of direct spike involvement.

Thank you for your comment. We agree with your opinion that cardiovascular manifestation of the COVID-19 mRNA vaccine in adolescents also raises this possibility of direct spike involvement. We have added this reference and made comment in Discussion of the revised manuscript (page 12, lines 300-302, ren font) as follows “Besides, a relatively higher incidence of myocarditis in adolescents after receiving COVID-19 mRNA vaccines (encoding S protein) also raises this possibility of direct S protein involvement in myocardial injury (Mansanguan S et al, 2022 Trop Med Infect Dis; Yonker LM et al., 2023 Circulation)”.

The above descriptions are the responses to your comments and suggestions.

Sincerely yours,

Yu-Hsun Kao, PhD

Graduate Institute of Clinical Medicine, Taipei Medical University